

# Trifluoroacetic acid deposition from emissions of HFO-1234yf in India, China, and the Middle East

Liji M. David[*1,2], Mary Barth[*3], Lena Höglund-Isaksson[4], Pallav Purohit[4], Guus J. M. Velders[5,6], Sam Glaser[1,7], and Akkihebbal. R. Ravishankara[*1,2]

[1]Department of Chemistry, Colorado State University, Fort Collins, CO 80523, USA
[2]Department of Atmospheric Science, Colorado State University, Fort Collins, CO 80523, USA
[3]Atmospheric Chemistry Observations and Modeling Laboratory, National Center for Atmospheric Research, Boulder, Colorado
[4]Air Quality and Greenhouse Gases Program, International Institute for Applied Systems Analysis (IIASA), Schlossplatz 1, 2361, Laxenburg, Austria
[5]National Institute for Public Health and the Environment (RIVM), PO Box 1, 3720 BA Bilthoven, The Netherlands
[6]Institute for Marine and Atmospheric Research Utrecht, Utrecht University, The Netherlands
[7]Currently at Tufts University, Medford, MA

*Address correspondence to:* liji.david@colostate.edu, barthm@ucar.edu, and a.r.ravishankara@colostate.edu

**Key points**

1. The expected concentrations of trifluoroacetic acid (TFA) from the degradation of HFO-1234 yf ($CF_3CF=CH_2$) emitted now and in the future by India, China, and the Middle East were calculated using GEOS-Chem and WRF-Chem models.
2. We conclude that, with the current knowledge of the effects of TFA on humans and ecosystems, the projected emissions through 2040 would not be detrimental.
3. We carried out various tests and conclude that the model results are robust.
4. The major uncertainty in the knowledge of the TFA concentrations and their spatial distributions is due to uncertainties in the future projected emissions.



**Abstract**

We have investigated trifluoroacetic acid (TFA) formation from emissions of HFO-1234yf, its dry and wet deposition, and rainwater concentration over India, China, and the Middle East with GEOS-Chem and WRF-Chem models. We estimated the TFA deposition and rainwater concentrations between 2020 and 2040 for four previously published HFO-1234yf emission scenarios to bound the possible levels of TFA. We evaluated the capability of GEOS-Chem to capture the wet deposition process by comparing calculated sulfate in rainwater with observations. Our calculated TFA amounts over the U.S., Europe, and China were comparable to those previously reported when normalized to the same emission. A significant proportion of TFA was found to be deposited outside the emission regions. The mean and the extremes of TFA rainwater concentrations calculated for the four emission scenarios from GEOS-Chem and WRF-Chem were orders of magnitude below the no observable effect concentration. The ecological and human health impacts now and continued use of HFO-1234yf in India, China, and the Middle East are estimated to be insignificant based on the current understanding, as summarized by Neale et al. (2021).

**Keywords:** HFO-1234yf, Trifluoroacetic acid, wet and dry deposition, India, China, the Middle East.

## 1. Introduction

The use of olefinic hydrofluorocarbons (HFCs) as substitutes for HFC-134a (1,1,1,2-tetrafluoroethane, $CF_3CFH_2$) are increasing in both the developed and developing countries (Velders et al., 2009). HFC-134a is a replacement for chlorofluorocarbons (CFCs) and hydrochlorofluorocarbons (HCFCs), which were phased out under the Montreal Protocol and its many amendments and adjustments (World Meteorological Organization, 2007). HFC-134a is a potent greenhouse gas with a 100-year global warming potential (GWP) of 1300 (Intergovernmental Panel on Climate Change (IPCC) report (Myhre et al., 2013)). HFO-1234yf (2,3,3,3-tetrafluoropropene, $CF_3CF=CH_2$) with a 100-year GWP of <1 (IPCC report (Myhre et al., 2013)) is a replacement for HFC-134a in automobile air conditioners (MAC) (Papadimitriou et al., 2008). The atmospheric degradation of HFO-1234yf leads to trifluoro acetyl fluoride ($CF_3C(O)F$) (Young and Mabury, 2010). $CF_3C(O)F$ hydrolyzes rapidly to yield trifluoroacetic acid (TFA, $CF_3-C(O)OH$), which is removed from the atmosphere by dry and wet deposition (George et al., 1994). The chemical lifetime of HFC-134a (~14 years) is such that it is reasonably well-mixed globally upon emission into the atmosphere. Therefore, its degradation and the TFA formed will occur across the globe. Only about 30% of the emitted HFC-134a leads to TFA (Kotamarthi et al., 1998). A large fraction of the formed TFA is deposited into the oceans. The fraction of HFC-134a degraded per year from one year's emission would be small, leading to small TFA in rainwater concentrations at a given location. However, as HFC-134a accumulates in the atmosphere, more TFA would be produced. HFO-1234yf has a shorter chemical lifetime of a few (~10) days (Myhre



et al., 2013) and its degradation leads almost exclusively (~100%) to $CF_3C(O)F$. Therefore, TFA
deposition per year of emission will be higher, depend on the year, and more localized spatially.
Previous studies have focused on TFA formation from emissions of either HFC-134a at
the current or previous levels (Kanakidou et al., 1995; Kotamarthi et al., 1998) or HFO-1234yf
substituted for current levels of HFC-134a usage (Luecken et al., 2010); then, they have mostly
scaled it for scenarios of HFO-1234yf emissions in the future over the continental U.S. and Europe
(Henne et al., 2012; Papasavva et al., 2009). Some works have distinguished between uses of HFC-
134a in MAC versus total usage, while others have evaluated maximum use scenarios. These
studies suggest that toxic levels of TFA in water bodies are not produced over Europe, North
America, and China if HFO-1234yf replaces all the current use of HFC-134a (Henne et al., 2012;
Kazil et al., 2014; Luecken et al., 2010; Wang et al., 2018). Russell et al. (2012) conducted a model
study to determine TFA concentration in terminal water bodies in the contiguous U.S., with TFA
deposition rates from Luecken et al. (2010). They found that after 50 years of continuous
emissions, aquatic concentrations of 1 to 15 $\mu g\ L^{-1}$ are projected, with extreme concentrations of
up to 50 to 200 $\mu g\ L^{-1}$ in the arid southwestern U.S.
Kazil et al. (2014) investigated, using the WRF-Chem model, the atmospheric turnover
time of HFO-1234yf, the dry and wet deposition of TFA, and the TFA rainwater concentration
over the contiguous U.S. between May and September 2006. They also examined where TFA
deposited emissions of three specific regions in the U.S. They concluded that the average TFA
rainwater concentration was 0.89 $\mu g\ L^{-1}$ for the contiguous U.S. Although Kazil et al. (2014) used
emission twice as large as that used by Luecken et al. (2010), the TFA rainwater concentrations
were comparable. Kazil et al. (2014) used the measured HFC-134a to CO ratio from the Los
Angeles area to obtain potential HFO-1234yf emissions. They also showed that TFA rainwater
concentrations reached significantly higher values (7.8 $\mu g\ L^{-1}$) at locations with very low
precipitation on shorter time scales. A comparably low TFA wet deposition occurred in the dry
western U.S. The work of Wang et al. (2018) is similar to that of Henne et al. (2012) and used the
GEOS-Chem model and examined the rainwater content and deposited amounts of TFA over
Europe, the U.S., and China, with similar findings. Henne et al. (2012) is the only study that used
two different models (FLEXPART and STOCHEM) to study the TFA deposition and rainwater
concentration over Europe.
The above-noted studies focused on the U.S. and Europe, and most recently China. The
U.S. and Europe emissions of the sum of HFC-134a and HFO-1234yf are expected to increase
only in proportion to the population in the future since the per capita number of MAC, stationary
AC, and other cooling units are unlikely to increase rapidly. India, China, and the Middle East are
the regions with expected large increases in HFO-1234yf use. In these regions, the number of units
and associated usage will increase rapidly as the economies grow. Perhaps Latin America and parts
of Africa would also see similar increases. The above-noted studies from the U.S. and Europe do
not allow us to draw firm conclusions about TFA's formation from realistic future emissions from
Asia (China and the Indian subcontinent), where the markets are not saturated, and meteorology
is very different from North America and Europe. The rate of degradation of parent compounds





and precipitation will differ in the warmer tropical and subtropical regions from those seen for the
U.S. and Europe; the seasonality will also be different. The precipitation across Asia is associated
with the Asian monsoon, which is stronger in comparison to the monsoon in southwest U.S. It is
also essential to look at the Middle East emissions since the studies of Kazil et al. (2014) and
Russell et al. (2012) showed that TFA rainwater concentrations are larger over drier areas of the
U.S., and there can be more accumulation in arid regions.
In 2019 the Kigali Amendment to the Montreal Protocol went into force. According to the
amendment the production and use of HFCs has to be phased down in the coming decades. This
should reduce the emissions of HFCs such as HFC-134a, but will likely increase emissions of
HFO-1234yf.
A description of the models used (GEOS-Chem and WRF-Chem), HFO-1234yf emission
scenarios, and the chemical scheme are given in section 2. In section 3, we compare the
precipitation in GEOS-Chem and WRF-Chem with observations. We evaluate the GEOS-Chem
model's ability to reproduce wet deposition by comparing sulfate rainwater concentrations with
observations. We performed simulations using both the models for the three domains individually
to calculate the TFA's dry and wet deposition and rainwater concentrations over India, China, and
the Middle East. We performed two-year runs in GEOS-Chem to check for interannual variability.
We compare our simulation results with other studies for the U.S., Europe, and China. The HFO-
1234yf emissions from all the regions together were also simulated to assess the interregional
effects. Major findings from this study are summarized in section 4.

**2.  Methods**
**2.1.  Model description**
*GEOS-Chem*: We used the GEOS-Chem (v12.0.3, www.geos-chem.org) global three-
dimensional chemical transport model driven by GEOS-FP assimilated meteorological data.
GEOS-Chem has a fully coupled tropospheric $NO_x$-$O_x$-hydrocarbon-aerosol chemistry. The
simulations were made at 2°×2.5° resolution and 47 vertical levels from the surface to ~80 km.
The wet deposition of aerosols and soluble gases by precipitation includes the scavenging in
convective updrafts, in-cloud rainout, and below-cloud washout (Amos et al., 2012; Liu et al.,
2001). The dry deposition was calculated using a resistance-in-series parameterization, which is
dependent on environmental variables and lookup table values (Wesely, 1989). The simulations
were conducted for 2015 and 2016 following a 2-month spin-up.
The global anthropogenic emissions were from Emissions Database for Global
Atmospheric Research (version 4.3). The global emissions are superseded by regional emission
inventories for India (Speciated Multi-pOllutants Generator (SMOG) and MIX) (Li et al., 2017;
Pandey et al., 2014; Sadavarte and Venkataraman, 2014), China (MIX), Europe (EMEP), U.S.
(National Emissions Inventory (NEI) 2011) (NEI2011, http://ww.epa.gov/air-emissions-
inventories), Canada (Criteria Air Contaminants (CAC), http://www.ec.gc.ca/), and Mexico
(BRAVO) (Kuhns et al., 2005). We used the biomass burning from Global Fire Emissions
Database (GFED) version 4 (Giglio et al., 2013). The biogenic VOC emissions were from the





Model of Emissions of Gases and Aerosols from Nature (MEGAN) version 2.1 inventory of
Guenther et al (2012). The details on other emissions are described in David et al. (2018, 2019).
**WRF-Chem:** The Weather Research and Forecast with Chemistry (WRF-Chem) model (Fast et
al., 2006; Grell et al., 2005) version 4.1.3, was used to simulate meteorology and chemistry over
India, China, and the Middle East individually. The WRF-Chem simulations were integrated for
14 months, beginning 1 November 2014 and ending 31 December 2015, with the first two months
of the simulation was used to spin up the model chemistry. The three model domains, shown in
Figure 1, have a horizontal grid spacing of 30 km and 40 vertical levels reaching a model top of
50 hPa. The vertical levels stretch in size with a fine resolution near the surface and a coarser
resolution in the upper troposphere. The model meteorology was initialized with Global Forecast
System (GFS) archived at 0.5° and a temporal resolution of 6 hours. Observational nudging is
applied for temperature, moisture, and winds to keep large-scale features in line with the observed
meteorology. The model physics and chemistry options that were used are summarized in Table
S1 in the supplementary information. The Model for Ozone and Related chemical Tracers
(MOZART) gas-phase chemical mechanism and the Global Ozone Chemistry Aerosol Radiation
and Transport (GOCART) scheme for aerosols (MOZCART) (Pfister et al., 2011) were used to
simulate ozone and aerosol chemistry. TFA chemistry was added to this chemical option. Six-
hourly results from the Community Atmosphere Model with Chemistry (CAM-Chem), which has
a similar chemistry mechanism as the WRF-Chem model configuration, were used (Tilmes et al.,
2015) to initialize trace gas and aerosol mixing ratios as well as to provide lateral boundary
conditions. HFO-1234yf and TFA were initialized with the GEOS-Chem results described above.
The Model of Emissions of Gases and Aerosols from Nature (MEGAN v2.04; Guenther, 2007)
was used to represent the net biogenic emissions for both gases and aerosols. Anthropogenic
emissions were from the Emissions Database for Global Atmospheric Research – Hemispheric
Transport of Air Pollution (EDGAR-HTAP) emission inventory (Janssens-Maenhout et al., 2015).
The Fire Inventory from NCAR version 1 (FINNv1.6; Wiedinmyer et al., 2011) was implemented
to provide daily varying emissions of trace species from biomass burning.
The wet removal scheme in WRF-Chem for MOZART chemistry, based on Neu and
Prather (2012), was used to compute the dissolution of soluble trace gases into precipitation and
their release into the gas phase upon evaporation of hydrometeors. Neu and Prather (2012) estimate
trace gas removal by multiplying the effective Henry's law equilibrium aqueous concentration by
the net precipitation formation (conversion of cloud water to precipitation, minus evaporation of
precipitation). Dry deposition of trace gases was described with the Wesely (1989)
parameterization. Diagnostic information on the wet and dry deposition of TFA was determined
every time step and accumulated values were included in the output files.

**2.2. Emissions**
HFO-1234yf is just now entering the market driven by regional (e.g., the European Union's
MAC Directive 2006/40/EC), national (e.g., Japan and U.S.) F-gas regulations, and the Kigali
Amendment to the Montreal Protocol. HFC-134a is currently the primary working fluid of MAC





and other applications (refrigerant, insulating foams, and aerosol propellants). Therefore, we have
to estimate the future emission levels from the three regions of interest. Unlike the developed
countries, India, China, and the Middle East are growing rapidly and the use of air conditioning
and refrigeration (and other uses of HFCs and HFOs) are expected to increase rapidly. Therefore,
one has to consider the likely economic growth and other factors in estimating emissions levels.
Here, we explore a few different potential scenarios for emissions of HFO-1234yf.
TFA production from HFO-1234yf increases linearly with the rise in HFO-1234yf
emissions, i.e., there is no feedback on this process since the primary drivers for the degradation
of this chemical, the OH radical, will not be altered by their relatively small emissions. In addition,
the changes in the abundance of OH in the troposphere in the next few decades are unlikely to be
different (say <13%) than the current levels based on the changes seen over the past few decades
(Rigby et al., 2017). Therefore, we can estimate the extent of TFA formation from a set of
modeling calculations that employed a fixed total amount of HFO-1234yf from each region. After
that, we can calculate the extent of TFA formation for various possible emission scenarios.
We used four future HFO-1234yf emissions scenarios for the 2020 to 2040 period: (1)
estimate of the upper range scenario HFO-1234yf emissions based on Velders et al. (2015)
estimate; (2) lower range scenario of HFO-1234yf based on Velders et al. (2015) estimate; (3) the
Greenhouse Gas Air Pollution Interactions and Synergies (GAINS) model (Amann et al., 2011)
with maximum technically feasible reduction (MTFR) estimates of HFO-1234yf; and (4) the
GAINS 'maximum HFO' (max HFO) scenario. Given the relatively short lifetime of HFO-1234yf,
the TFA production per year is dependent only on the emissions in that year. Figure 2 shows the
HFO-1234yf emission projection from India, China, and the Middle East for the four scenarios
between 2020 and 2040. The scenario based on Kumar et al. (2018) is similar to the fourth scenario
we considered. Therefore, we have not specifically included this possibility.
The emission estimates of HFO-1234yf in the GAINS model depends on when countries
will comply with the Kigali Amendment, current and future emissions based on country-level
activity data, uncontrolled emission factors, the removal efficiency of emission control measures,
and the extent to which such measures are applied (Purohit and Höglund-Isaksson, 2017). The
GAINS model uses the fuel input for the transport sector that is provided by the exogenous
projections (e.g., International Energy Agency's World Energy Outlook 2017). First, using the
annual mileage per vehicle (veh-km) and specific fuel consumption (SFC), GAINS estimates the
number of vehicles (by type, fuel). Second, using the penetration rate of a MAC, the number of
vehicles with MAC is calculated. Next, using the specific refrigerant charge (different for MAC
used in vehicle types), the HFO-1234yf consumption in mobile air conditioners is calculated. Note
that the HFO-1234yf is assumed to be substituted for HFC-134a, one-to-one, in all vehicles. For
HFO-1234yf use in MAC, HFO-1234yf emissions are estimated separately for "banked"
emissions, i.e., leakage from equipment in use, and for "scrapping" emissions, i.e., emissions
released at the end-of-life of the equipment. The leakage rate in the GAINS model assumes a
percentage of the charge per year. For example, if the refrigerant charge in MAC is 0.5 kg then the
emissions from the bank will be 0.05 kg (= 0.5 kg×0.1) per year, where the leakage rate is 10%





per year. This leakage rate is a steady refrigerant loss through seals, hoses, connections, valves,
etc. from every MAC over the entire use-phase (annually). At the end-of-life, the scrapped
equipment is assumed to be fully loaded with refrigerant, which needs recovery, recycling, or
destruction. At the same time, if there are regulations in place (e.g., MAC Directive 2006/40/EC
in European Union) - a package of measures including leak prevention during use and refill,
maintenance, and end of life recovery, and recollection of refrigerants, GAINS consider these good
practices as a control option with a removal efficiency of 50% for *in-use* and 80% for *end-of-life* -
based on secondary sources (Purohit et al., 2020). However, no such measures are assumed for
India, China, and the Middle East. Thus, these emissions can be considered the maximum likely
emissions. The MTFR version of the GAINS scenario assumes that the maximum technically
feasible reductions are applied across the sectors in India, China, and the Middle East. The Velders
et al. (2015) emissions also follow the Kigali amendment. The 'Shared Socioeconomic Pathways'
(SSPs) SSP3 and SSP5 are the lower and upper range scenarios, respectively, used in Velders et
al. (2015) calculated for 11 geographic regions and 13 use categories. Kumar et al. (2018) highlight
that many applications in India will likely transition to something other than HFOs. These trends
are not unique to India and will likely be replicated in China and the Middle East. Therefore, the
four HFO-1234yf emission scenarios for the 2020 to 2040 period represent emissions that are
higher than should be expected and, therefore, are upper limit estimates of the potential impact of
TFA in these regions.
To minimize numerical errors (in using small emissions) and compare them with previous
studies, we used HFO-1234yf emissions of ~40 Gg yr$^{-1}$ from each of these regions. This value
corresponds to 2025 projected emissions from the GAINS model over India and the Middle East
if all the applications were to use HFO-1234yf in place of HFC-134a for China, the emission values
are for 2016 from Wang et al. (2018). Figure S1 in the supplementary information shows the annual
spatial distribution of HFO-1234yf over India, China, and the Middle East as simulated in: (i)
GEOS-Chem; and (ii) WRF-Chem models. We distributed this total emission across the
country/region of interest by scaling the emission to known anthropogenic CO emissions used in
the model. The anthropogenic CO is a good tracer for HFO-1234yf emissions since they originate
from similar applications (especially the transport sector) and in proportion to the distribution of
economic activities in the region/country of interest. We show the total emissions in each of the
three regions in both the models in the figure. The distribution varies to a small extent with the
season (shown in Figure S2 in the supplementary information), and the monthly variation in
emission is similar in both models. We also simulated GEOS-Chem over the US and Europe using
the total HFO-1234yf emissions from Wang et al. (2018) and Henne et al. (2012), respectively.

**2.3.    Chemical scheme**
The chemical degradation of HFO-1234yf and the production of TFA were added to both
the GEOS-Chem $NO_x$-$O_x$-hydrocarbon-aerosol chemistry scheme and the WRF-Chem
MOZCART chemistry scheme. The detailed chemical scheme for the formation of TFA from
HFO-1234yf is shown in Burkholder et al. (2015) and, therefore, not repeated here. The simplified
representation of TFA production follows that of Kazil et al. (2014):
$$OH + CF_3CF = CH_2 \rightarrow CF_3C(O)F$$
$$CF_3C(O)F \rightarrow inclouds \rightarrow CF_3C(O)OH$$
$$CF_3C(O)F(clouds) \rightarrow Wet\ depostion$$
$$CF_3C(O)F(clouds) \rightarrow CF_3C(O)OH(gas\ phase)$$
$$CF_3C(O)OH(gas\ phase) + OH \rightarrow loss$$
$$CF_3C(O)OH(gas\ phase) \rightarrow dry\ deposition$$
The conversion of HFO-1234yf to TFA includes OH-initiated reaction of $CF_3CF=CH_2$ with
a temperature-dependent rate coefficient of $1.26\times10^{-12}$ exp(-35/T) cm$^3$ molecule$^{-1}$ s$^{-1}$ to produce
gas-phase $CF_3C(O)F$ (note that the initial OH reaction is the rate-limiting step in the conversion).
The gas-phase removal of $CF_3C(O)F$ is not rapid but hydrolyzes to TFA in water (George et al.,
1994). The hydrolysis process is added to the heterogeneous chemistry with a hydrolysis rate of
150 s$^{-1}$, and Henry's Law solubility constant of 3 M atm$^{-1}$. TFA is highly soluble in cloud water.
Upon cloud evaporation, the dissolved TFA is released into the gas phase. The gas-phase TFA is
expected to be deposited either via dry or wet deposition using Henry's Law solubility constant of
$9\times10^3$ M atm$^{-1}$ at a standard temperature of 298.15 K, $\Delta H/R$ = 9000 K, dissociation coefficient of
0.65 at 298.15 K, and $\Delta E/R$ = -1562 K. Thus, at cloud temperatures (generally <290 K) the
effective Henry's Law of TFA is high, characterizing TFA as a highly soluble gas. The dry
deposition rate for TFA is assumed to be the same as that for nitric acid (Henne et al., 2012; Kazil
et al., 2014; Luecken et al., 2010). We also included the potential loss of gas-phase TFA by its
reaction with OH radicals with a rate coefficient of $9.35\times10^{-14}$ cm$^3$ molecule$^{-1}$ s$^{-1}$. Other potential
losses of HFO-1234yf via reaction with Cl, O$_3$, and NO$_3$ are very small and all yield the same set
of products. Therefore, we have not included them in the model. We also examined the possible
removal of gas phase TFA by its reaction with Criegee intermediate (CI). We used the Bristol
group's calculated concentrations of the Criegee intermediates (Khan et al., 2018).
**3.    Results and Discussion**
**3.1.    Sulfate concentration in rainwater**
Wet deposition is one of the primary removal processes for TFA. This deposition depends
on precipitation amounts and how well our model captures the wet deposition process, making it
crucial to evaluate the models used here to capture these two factors.
First, we compared the annual total precipitation amounts calculated by GEOS-Chem and
WRF-Chem with the observed daily total accumulated precipitation from the Tropical Rainfall
Monitoring Mission (TRMM_3B42_daily product) in the three regions (Figure S3a in the
supplementary information). The TRMM product is at a 0.25°×0.25° resolution. The spatial
distribution of seasonal total precipitation in the three domains from the two models and TRMM
is shown in Figure S4 in the supplementary information. Both the models captured the seasonal



precipitation patterns. As seen in Figure S3a, the total precipitation amounts were a factor of 1.5-
2 higher in GEOS-Chem compared to WRF and TRMM. (The ratio of total precipitation between
GEOS-Chem and WRF-Chem(TRMM) were 2.6 (1.5), 2.2 (1.5), and 2.2 (1.4) for India, China,
and the Middle East, respectively). WRF-Chem underestimated the precipitation amounts
compared to TRMM in the three regions. Kumar et al. (2012; 2018) have addressed the
precipitation biases in WRF-Chem compared to TRMM over South Asia. We attribute the higher
precipitation in GEOS-Chem to: (a) the different model physics used; (b) the effects of a
meteorology-driven chemistry transport model (GEOS-Chem) versus an "online" chemistry
transport model (WRF-Chem) where chemistry is solved at the same time step as the meteorology;
(c) and to the different grid spacings used by the two models, noting that the coarse GEOS-Chem
grid cells contain several convective storms compared to that in WRF-Chem. The monthly
variation in total precipitation is shown in Figure S3b (supplementary information), and both
models have similar trends as that observed by TRMM.

To evaluate the accuracy of the TFA wet deposition, it is useful to compare sulfate wet
deposition amounts produced by the oxidation of $SO_2$. The emissions of $SO_2$ are comparable in
both the models (shown in Figure S5 in the supplementary information). We have measurements
of sulfate rainwater concentrations in some of the regions. Further, the lifetime of $SO_2$ in the
troposphere is comparable to that of HFO-1234yf. We hasten to add that while the HFO-1234yf
degradation is controlled by gas phase OH reactions, that of $SO_2$ include both gas and condensed
phase processes.  However, the removal of both sulfate and TFA are due to condensed phase
reactions. The WRF-Chem model has been shown to capture the sulfate rainwater concentration
over the continental U.S. by Kazil et al. (2014); we expect it to do well over this study's regions.
However, GEOS-Chem has not been evaluated previously. There are no networks for measuring
sulfate rainwater concentration in India and the Middle East. Yet, there are some observations of
rainwater sulfate in the published articles in all three domains. The available data are sparse, and
the observations for 2015 (the modeled year) are even fewer to make a comparison with WRF-
Chem simulations. However, GEOS-Chem simulations were available from our previous work for
2000-2015. We used those results to compare with observations during that period. The
observation locations (over land only) in the three domains are shown in Figure S6 in the
supplementary information. Figure 3 shows the scatter plot of simulated and observed sulfate
rainwater concentration in the three domains. Table 1 lists the statistics of the comparison between
GEOS-Chem and the observations. Rainwater sulfate amounts calculated by GEOS-Chem
correlate well (R>0.80) with observations. We see a bias of -13, -13, and -3% in India, China, and
the Middle East domains, respectively. The negative bias in GEOS-Chem sulfate rainwater
concentration could be because the model integrates over a large area while the observations are
point locations. It could also be, as noted earlier, because GEOS-Chem yields higher amounts of
precipitation and thus could lead to smaller rainwater concentrations. We suggest that these values
are good to at least a factor of two. In summary, the GEOS-Chem model shows considerable skill
in reproducing mean sulfate rainwater concentrations and spatial variability of sulfate rainwater
concentrations; therefore, it can be utilized to calculate TFA wet deposition.



**3.2. Comparison of calculated TFA with previous studies**
Before presenting the results of the calculations for India, China, and the Middle East from
the present study, we note that our models agree with the previous studies over the U.S. (Kazil et
al., 2014; Luecken et al., 2010), China (Wang et al., 2018), and Europe (Henne et al., 2012). Figure
4 shows the comparison of annual mean (a) TFA deposition (dry and wet combined), and (b) TFA
rainwater concentration over the U.S., China, and Europe. We have normalized the emissions to
match those of the previous studies for meaningful comparisons. (The emissions used to compare
TFA from the U.S., China, and Europe are 24.5, 42.7, and 19.2 Gg yr$^{-1}$, respectively.) We note that
average deposition rates differ in the model because of the differences in calculations' domain
sizes. Given that the models vary in their versions, meteorology, physics, and the expected model
variabilities, the observed agreement is reasonable. We also show the comparison with our
calculations over the U.S. for the summer months with previous studies (Kazil et al., 2014;
Luecken et al., 2010; Wang et al., 2018) (Figure S7 in the supplementary information).
**3.3. Atmospheric mixing ratios**
Figure 5 shows the annual mean mixing ratios of HFO-1234yf over India, China, and the
Middle East as simulated by GEOS-Chem and WRF-Chem. We present here only results with
emissions in GEOS-Chem(WRF-Chem) of 41.3(41.9), 40.6(39.9), and 37.8(38.1) Gg yr$^{-1}$ from
India, China, and the Middle East, respectively. Expected TFA for other emissions can be simply
scaled to the emissions of interest. The annual mean mixing ratio of HFO-1234yf in India, China,
and the Middle East as simulated by GEOS-Chem(WRF-Chem) were 2.87(3.94) ppt, 2.49(3.70)
ppt, and 1.82(2.49) ppt, respectively, and below 1 ppt (as seen in GEOS-Chem) outside of the three
regions. The annual mean mixing ratio in the China domain was comparable to Wang et al. (2018).
The highest (>40 ppt) simulated annual mean HFO-1234yf mixing ratio for India was in the Indo-
Gangetic Plain (IGP), for China in the northeast region, and for the Middle East in northern Iran.
The emission hotspots (Figure S1 in the supplementary information) in the three regions led to the
largest annual mean HFO-1234yf mixing ratios in those regions. The WRF-Chem simulated higher
annual mean HFO-1234yf mixing ratios compared to GEOS-Chem. Differences in annual mean
HFO-1234yf mixing ratios between models for the same amount of emissions have been reported
also by Henne et al. (2012). However, the overall spatial patterns are comparable between GEOS-
Chem and WRF-Chem. It should be noted that the change of HFO-1234yf emissions in any of the
three regions would change the HFO-1234yf mixing ratio within that region and will have minimal
effect on other regions.
**3.4. TFA deposition**
GEOS-Chem simulated mean total deposition rates (dry and wet deposition combined) to
be 0.874, 0.501, and 0.477 kg km$^{-2}$ yr$^{-1}$, respectively, in India, China, and the Middle East domains
for emissions of 41.3, 40.6, and 37.8 Gg yr$^{-1}$, respectively. WRF-Chem simulated mean deposition
rates (dry and wet) were 0.802, 0.342, and 0.284 kg km$^{-2}$ yr$^{-1}$ in India, China, and the Middle East





domains, respectively (Figure S8 in the supplementary information). Figure 6 shows the annual
total dry and wet TFA deposition rates in the three domains. The total annual dry deposition in
GEOS-Chem and WRF-Chem over the India domain was largest in eastern India and Bangladesh,
reaching up to 2 kg km$^{-2}$ yr$^{-1}$. The wet deposition in the India domain mostly occurred in the
Himalayas' foothills, eastern IGP, parts of central India, and southwest India and was >3.5 kg km$^2$
yr$^{-1}$. In the China domain, the total dry and wet deposition rates in GEOS-Chem and WRF-Chem
were highest in southeast China. The total dry deposition rate in the Middle East domain was
highest in northern Iran. The wet deposition rate was the largest in parts of Iran, with differences
between the models. The wet deposition dominated the total TFA deposition. The combined annual
total deposition pattern was similar to that of wet deposition in the three domains (Figures 6 and
S8 in the supplementary information). The seasonal total deposition rates of TFA from dry and
wet depositions in the three domains are shown in Figure S9 in the supplementary information.
The seasonal deposition rates were highest for June-September, June-August, and April-October
in India, China, and the Middle East domains, respectively.
Figure 7 shows the percentage contribution of dry and wet deposition to total TFA
deposition between GEOS-Chem and WRF-Chem in the three domains. It should be noted that the
sum of the two (dry and wet) percent contribution do not add up to exactly 100% because of
transport in and/or out of the domains. For the total amount of HFO-1234yf emissions mentioned
in Figure S1(supplementary information) and discussed in section 2.2, the total TFA deposition
(dry and wet combined) in India, China, and the Middle East domains from GEOS-Chem were
23.4, 20.5, and 18.7 Gg yr$^{-1}$, respectively. The total annual dry(wet) deposition amounts account
for 21(36)%, 20(31)%, and 20(29)% of the annual emissions of HFO-1234yf in GEOS-Chem. In
WRF-Chem, annual total TFA deposition was 19.4, 12.1, and 9.9 Gg yr$^{-1}$, respectively, in India,
China, and the Middle East domains. The dry(wet) TFA deposition was 10(37)%, 3(23)%, and
4(26)% of the emissions in India, China, and the Middle East domains, respectively. Table S2
(Supplementary Information) shows the seasonal TFA deposition (dry and wet) calculated from
GEOS-Chem and WRF-Chem models in the three domains. The lower TFA deposition in WRF-
Chem compared to GEOS-Chem is due to the venting of surface emissions into the free
troposphere (Grell et al., 2004; Kazil et al., 2014) that leads to lower dry deposition in WRF-Chem
(Figure 7a). The differences in deposition between models can also be attributed to differences in
model resolutions, model transport, meteorological conditions (e.g., precipitation), and cloud
treatment. These differences highlight the need for multi-model simulations to estimate the likely
variation in these parameters.
Figure 8 shows the total TFA deposition (dry and wet combined) for the four emission
scenarios (Figure 2) calculated from GEOS-Chem and WRF-Chem. Our results show that the
differences in the calculated extent of TFA formed and deposited are about a factor of two between
the models. In all cases, the computed TFA dry and wet deposition varies linearly with the
emissions. Therefore, we can calculate the amounts of TFA formed and deposited for any
envisioned emission of HFO-1234yf.



### 3.5. Rainwater concentrations

Figure 9 shows the monthly variation in mean TFA rainwater concentration in the three domains calculated from GEOS-Chem and WRF-Chem. The TFA rainwater concentration also varies linearly with the emissions. Figure 9 shows the following: (a) higher concentrations are to be expected when there is little rain/precipitation (Figure S3b in the supplementary information) to remove TFA. This point has been noted in previous studies (Kazil et al., 2014; Russell et al., 2012; Wang et al., 2018). So, if all the TFA were concentrated into a small amount of rain, the concentrations have to be larger. Such events are infrequent. They are, relative to the rainier regions, more frequent in the Middle East. The large rainwater concentration does not mean that the amount of deposited TFA is larger; (b) The rainwater concentrations varied inversely with the precipitation amount, as seen by comparing the rainwater TFA levels with the total precipitation (Figure S3 in the supplementary information). A clear signal for the rainfall variation was seen over India, where the monsoon season (June, July, August, and a part of September) bring large and almost constant precipitation. This large precipitation makes the TFA rainwater concentrations extremely small. In other words, this is simply a dilution effect; (c) When the rainfall is small, there are considerable variations as one would expect. Lesser total precipitation arises because of fewer showers and often in spatially and temporally sporadic events. So, the concentrations can vary a great deal. This was also evident over China during dry seasons; and (d) the calculated TFA rainwater concentrations were comparable to previous calculations for China (scaled to emissions, Figure 4b).

It is important to know the regions of high TFA rainwater concentrations. Therefore, we plotted the spatial pattern of annual mean TFA rainwater concentration in the three domains from both models (Figure S10 in the supplementary information). It is noticeable that most of the regions in all three domains did not have high TFA rainwater concentrations. There were some grids with TFA rainwater concentrations that exceeded 50 µg L$^{-1}$ for emissions of ~40 Gg yr$^{-1}$. The high TFA rainwater concentration seen in the western part of India and China domains is because of input at the lateral boundaries from a global model. As mentioned in section 3.1, the precipitation in GEOS-Chem was higher, resulting in lower TFA rainwater concentration. Focusing on the highest possible rainwater concentrations is misleading since that does not tell us the amount of wet TFA deposition, which is shown in Figure 6. However, it is clear that if the emissions of HFO-1234yf reach the large numbers noted by the IIASA/GAINS model (max HFO, Figure 2d) for 2040, there will be significant areas with larger TFA rainwater concentrations. The wet deposition does not tell the whole story either since a substantial fraction of the rainwater ends up in the oceans every year. The estimation of the TFA retained on land will be critical for further estimating the long-term impact. Such a hydrology study is warranted but beyond the scope of this work.

### 3.5.1. Comparison of expected TFA rainwater levels with No Observable Effects Concentrations

The primary reason for carrying out these calculations was to estimate the potential impact of HFO-1234yf usage in the three regions of the study for the current and future emissions. The effects of interest here are TFA formation from HFO-1234yf and its consequences to human and





ecosystem health. Figure 10 shows the mean TFA rainwater concentration for the four emission
scenarios calculated from GEOS-Chem and WRF-Chem. In all the scenarios, the annual mean
TFA rainwater concentration was well below the no observed effect concentration (NOEC) for
aquatic species, which is >10,000 µg L$^{-1}$ (Solomon et al., 2016), with an outlier for the most
sensitive alga as 120 µg L$^{-1}$ (Boutonnet et al., 2011).
Neale et al. (2021) have summarized the impact of TFA on human and ecosystem health.
Their conclusion suggests that the NOEC on aquatic systems is >10,000 µg L$^{-1}$. As shown in
Figures 9, 10, and S10 (supplementary information), the expected rainwater concentrations are at
least two orders of magnitude lower than the NOEC. Also, the rainwater concentrations of TFA,
even for the 2040 emissions, are roughly comparable to those currently observed in China (Chen
et al., 2019) and about ten times greater than those presently observed over Germany (Freeling et
al., 2020). They also note that large TFA concentrations have been observed in people's blood in
China with no ill effects on the endpoints measured in that work (Duan et al., 2020).
TFA quantities deposited via dry deposition to land and vegetation would be much smaller
than those noted in Neale et al. (2021) to have any significant detrimental health effect. Indeed,
they note that there are other sources of TFA that are much higher than those expected from HFO-
1234yf degradation. Neale et al. (2021) also point out that the TFA deposited to snow in the Arctic
would not significantly contribute to marine water bodies even if it all melted down since the
volume of the melt would be much smaller than those of the receiving water bodies.
Lastly, since TFA can accumulate over land and water bodies, we can estimate the
influence of accumulation on the potential future impacts. The total TFA amount in rainfall would
not change. However, the amounts in water bodies could increase. For the 20 years modeled here,
the total TFA in water bodies would be larger than those observed for 2020 if TFA merely
accumulates. It is hard to calculate precisely where the water bodies would accumulate TFA
without a hydrological model. However, these values would still be orders of magnitude smaller
than the NOEC of >10,000 µg L$^{-1}$. For example, if all the TFA produced in these regions were to
end up in the top 15 meters of the world's oceans, we expect the TFA levels to increase by about
0.015 µg L$^{-1}$ by 2040.
Based on these observations, and assuming that the NOEC concentration holds, it appears
that the TFA from the expected emissions of HFO-1234yf in these three regions would not
constitute a health threat to plants or humans (even if we assume that there is no water treatment
to remove TFA in drinking water).

**3.6.    Interannual variability**
The model results discussed in the previous sections are for one year, 2015. To assess the
influence of interannual variability in meteorology, we simulated the TFA deposition and
rainwater concentration for 2016 with the GEOS-Chem model for the total HFO-1234yf emissions
described in section 2.2. Figure 11 shows the fraction of TFA in the three domains for 2015 and
2016 that is: (a) dry deposited; (b) wet deposited; and (c) the annual mean TFA rainwater
concentrations. The total precipitation in both years was comparable (shown in Figure S11 in the



supplementary information). The results of our two-year simulations lead us to conclude that the
interannual differences are small. Therefore, we suggest that the results of 2015 are applicable
going forward to 2040.
**3.7.**      **Simultaneous emissions from multiple regions**
It is important to note that most TFA is deposited outside of the domains, even though the
estimated lifetime of HFO-1234yf is about ten days. Therefore, TFA is dispersed significantly
from the source region. Figure 12a shows that roughly 25-50% of the HFO-1234yf emitted from
a given region was converted and deposited (via dry and wet deposition) as TFA within the domain
(see Figure 1 for domain boundaries). Figure 12b shows the percentage of TFA deposition (dry
and wet combined) calculated from GEOS-Chem and WRF-Chem within the three domains over
land. The remaining TFA was transported outside the domain. It is difficult to quantify the exact
locations of these depositions outside the domain since the concentrations get very small even
though in the aggregate that accounts for somewhere between 30% and 45%. The fraction that was
deposited within the region of emission was even smaller and ranged between 7% and 27%.
Therefore, it can be concluded that a significant fraction ended up in the oceans. This is especially
true for India and the Middle East emissions. Interestingly, a substantial amount of the TFA from
the Middle East emissions deposited in the Arabian Sea. Therefore, we conclude that even though
HFO-1234yf is short-lived, it is still sufficiently long-lived to travel thousands of kilometers. Such
an expectation is in accord with the calculated distances traveled by an airmass for even about 2
m s[1].
The deposition outside of the region and domains also means that the emitting regions are
not the only area affected by their emission of HFO-1234yf. This is in spite of the relatively short
turnover time of HFO-1234yf. (Note: We call this the turnover time since because of the way we
calculate it in the model.) Since the three countries/regions studied here are adjacent to each other
and their domains overlap (Figure 1), it is possible to estimate the impact of the neighbors'
emission on each other. We consider the emissions over the rest of the world (excluding India,
China, the Middle East, the US, and Europe) from Fortems-Cheiney et al. (2015) assuming HFC-
134a is substituted with HFO-1234yf on a mole-per-mole basis (the maximum likely emissions
scenario). Figure S12 in the supplementary information shows the annual spatial distribution of
HFO-1234yf emissions from all the regions as simulated in GEOS-Chem. The percentage
deposition of TFA (dry and wet combined) from global and regional (individual regions) emissions
of HFO-1234yf is shown in Figure 13. The TFA deposition increased by 7-18% in the three
domains because of the emissions from its neighbors. Figure 13 suggests that if the entire world
switches to HFO-1234yf, the impact of TFA from the near and far neighbors would be noticeable,
but still be at most a factor of 2 or 3 larger. Figure S13 in the supplementary information shows
the spatial pattern of the annual total TFA via (a) dry and (b) wet deposition rates from global
emissions of HFO-1234yf. The dominant TFA deposition regions were most parts of India,
southeast China, parts of Iran, and the southern Arabian Sea. We discussed in section 3.5.1 the
potential impacts of such a global switch.



**3.8.    Reaction of TFA with Criegee intermediates**
We examined the influence of CI's potential reactions with TFA on its tropospheric levels.
We used the CI concentrations in the boundary layer (0-2 km) from Chhantyal-Pun et al. (2017)
in GEOS-Chem and simulated the model for seven months (January to July) using 2015
meteorology. Figure S14 in the supplementary information shows the mean surface CI
concentration for those seven months calculated at 2° x 2.5° spatial resolution. The CI
concentrations in the three regions of our study were less than 2500 molecules cm$^{-3}$. We calculated
the percentage decrease in total TFA deposition within the three domains by including the CI
chemistry. We assumed at all the CI reactions with TFA have the rate coefficient measured for
that of CH$_2$OO with TFA, i.e., $5 \times 10^{-18} T^2 e^{1620/T}$ cm$^3$ molecules$^{-1}$ s$^{-1}$. Figure 14 shows the
spatial pattern of decrease in total TFA deposition (dry and wet combined) for seven months. In
most of the locations within the three domains, the decrease in total TFA deposition was <2.5%.
At a few places in southeast Asia (Figure 14a), western China (Figure 14b), and northern Africa
(Figure 14c), the TFA deposition decreased by 7-25%. The decrease in TFA deposition due to CI
was 0.03, 0.32, and 0.08 Gg (total for seven months)  for India, China, and the Middle East
domains, respectively. Overall, the impact of CI on TFA deposition was small in the region of
study.
**4.    Summary**
We have investigated TFA formation from emissions of HFO-1234yf, its dry and wet
deposition, and rainwater concentration over India, China, and the Middle East with GEOS-Chem
and WRF-Chem models. We estimated the TFA deposition and rainwater concentrations between
2020 and 2040 for four HFO-1234yf emission scenarios. The models were simulated for a year
(2015), with additional 2016 simulations to understand the interannual variability. We also
simulated the model using global emissions to assess interregional effects on TFA deposition. The
main results of the study are summarized below:
• Using two models at different spatial resolutions helped us assess the variation in model
transport, precipitation, and cloud treatment. These variations yield slightly different
calculated TFA levels from the emission of HFO-1234yf. Even though there are discernable
differences, the overall conclusions are the same and point to this study's robustness.
• The accuracy of the GEOS-Chem model's ability to calculate wet deposition over the regions
of interest was tested by comparing calculated sulfate rainwater concentration with
observations. The model reproduces well the multiyear sulfate rainwater concentration (-3%
to -13% bias) and its spatial variability (R>0.80) in the three domains.
• Our calculated TFA amounts over the U.S., Europe, and China were comparable to those
previously reported when normalized to the same emissions.
• The controlling factor for the amount of TFA from HFO-1234yf is its emissions. The
uncertainties in the models and chemistry are secondary to the extent of emissions.



- The TFA deposition was largest over eastern India, southeast China, northern Iran, and the southern Arabian Sea. The TFA wet deposition was comparable between the two models.
- There are large variations in TFA rainwater concentrations associated with rainfall extent. The mean TFA rainwater concentration calculated for the four emission scenarios from GEOS-Chem and WRF-Chem was below the no observable effect concentration (NOEC), suggesting the ecological and human health impacts to be not significant.
- With a chemical turnover time of HFO-1234yf of 10 days, its impact is not local and extends well beyond the region of emissions. This study highlights the enhanced TFA formation by the simultaneous use of HFO-1234yf by neighboring regions. If all the Northern Hemisphere countries were to use HFO-1234yf, the impact would be higher by a factor of 2 or 3. However, these amounts are still much lower than the NOEC noted above.
- We estimate that continued use of HFO-1234yf in India, China, and the Middle East are unlikely to lead to detrimental human health effects based on the current understanding of the effects of TFA in water bodies, as summarized by Neale et al. (2021). (Note we do not assume the water is treated specifically to remove TFA before consumption.)
- We note that a hydrology model of the water flow and TFA concentrations in them would be beneficial to quantify the extent of TFA accumulation in pools and flow out to large water bodies.

**Acknowledgement:**
We are grateful to Jan Kazil (NOAA/CSL) and Rajesh Kumar (NCAR) for help with the WRF-Chem. We are thankful to Jared Brewer and Viral Shah for helping with TFA chemistry in GEOS-Chem. We are thankful to Kirpa Ram for providing the sulfate rainwater concentration data over India. We are grateful to Anwar Khan, Rabi Chhantyal Pun, Dudley Shallcross, and Andrew Orr-Ewing for providing their calculated Criegee intermediate concentrations. This work was funded by the Global Forum for Advanced Climate Technologies.

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





**Tables**
**Table 1.** The slope, correlation coefficient (R), intercept (c), mean bias (MB), and the number of
points (N) of simulated (GEOS-Chem) and observed sulfate rainwater concentration over the three
domains.

| Region | Slope | R | c | MB | N |
|---|---|---|---|---|---|
| India | 0.771 | 0.816 | 0.210 | -0.255±0.778 | 54 |
| China | 0.799 | 0.911 | 0.655 | -1.07±2.700 | 89 |
| Middle East | 1.42 | 0.880 | -2.81 | -0.187±2.71 | 5 |




**Figures**

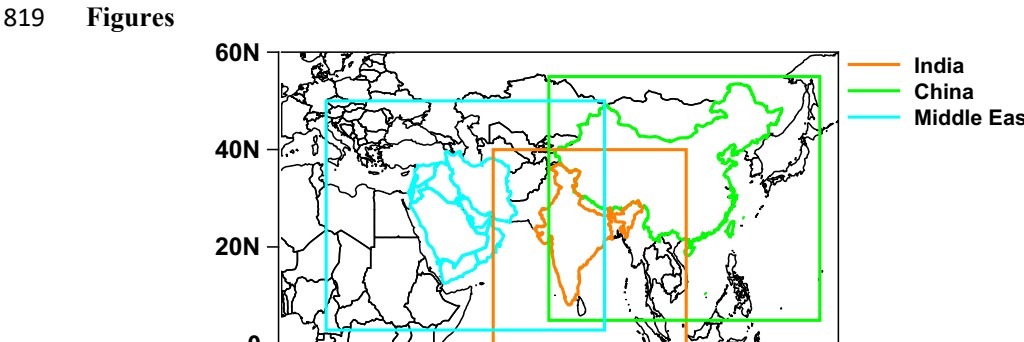

**Figure 1.** The model domains for India, China, and the Middle East used in the present study and also for WRF-Chem simulations. The land regions for the emissions are shown in color.

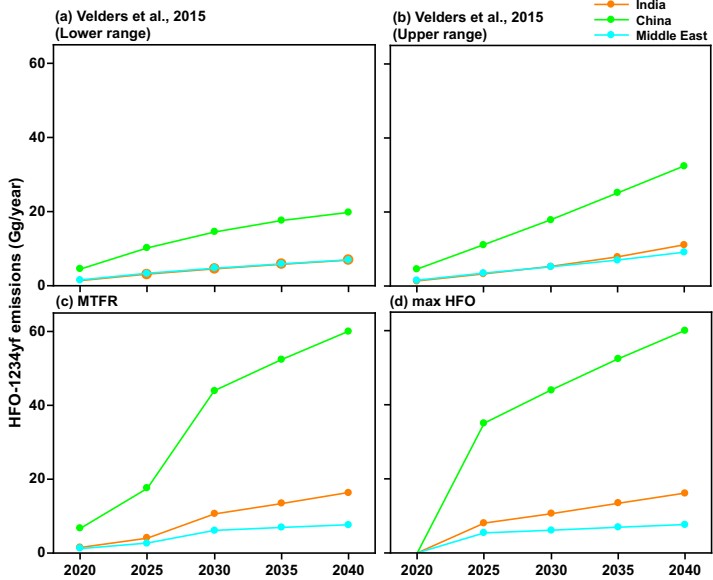

**Figure 2.** The projected HFO-1234yf emissions scenarios between 2020 and 2040 from Velders et al. (2015) (a) lower and (b) upper ranges, IIASA GAINS model for (c) Maximum Technically Feasible Reduction (MTFR) and (d) max HFO in India, China, and the Middle East.



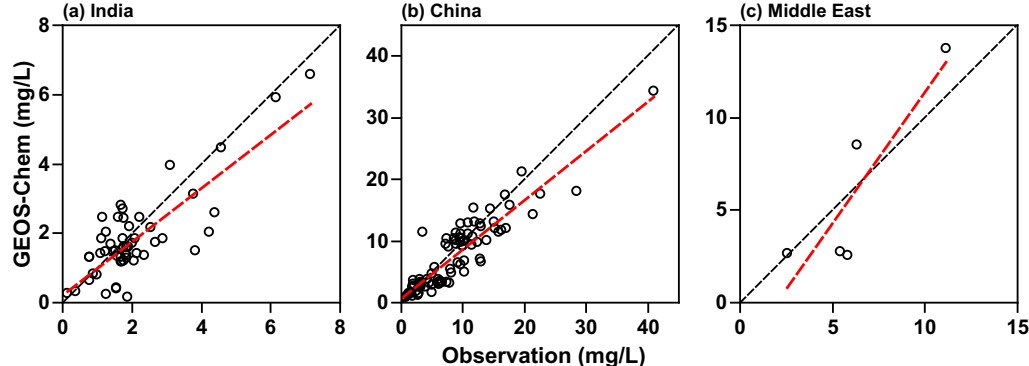

**Figure 3.** Scatter plot of simulated and observed sulfate rainwater concentration in (a) India, (b) China, and (c) the Middle East for 2000-2015. The linear regression line is shown in red. The black dashed line corresponds to slope = 1. The data for the Middle East are very limited.

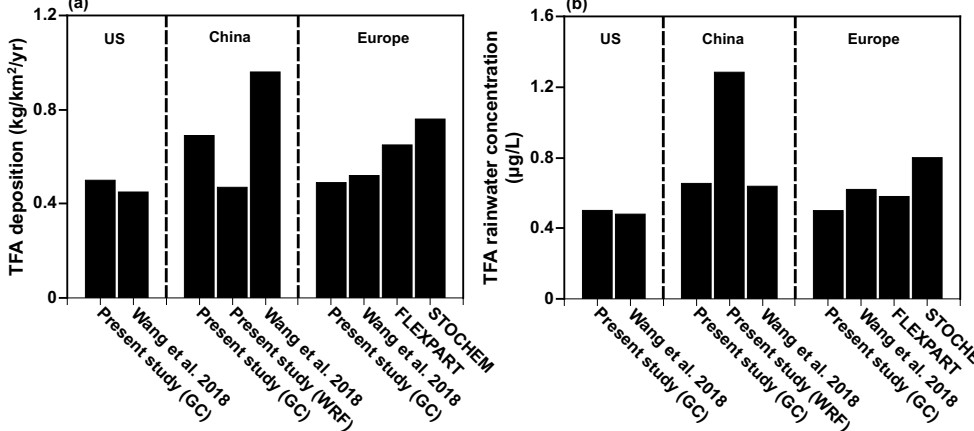

**Figure 4.** Comparison of the present study with other studies over the U.S., China, and Europe for (a) TFA deposition, and (b) TFA rainwater concentration. Note the emissions are 24.53 Gg yr$^{-1}$, 42.65 Gg yr$^{-1}$, and 19.16 Gg yr$^{-1}$ for the U.S., China, and Europe, respectively.




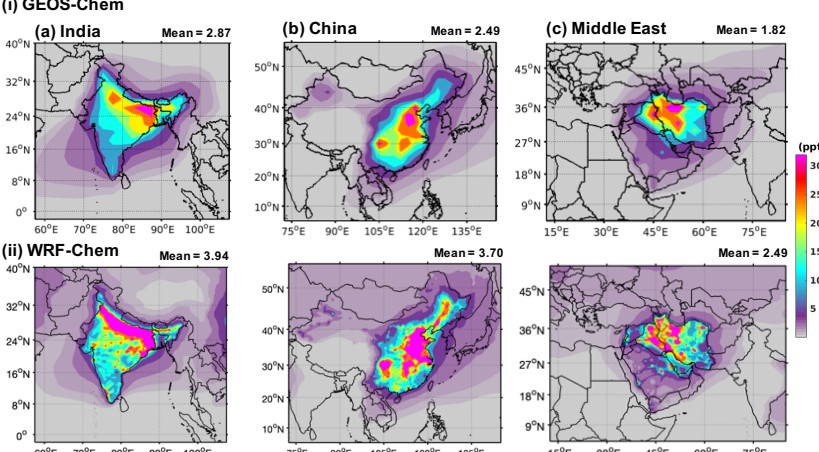

**Figure 5.** Annual mean surface mixing ratios of HFO-1234yf simulated in (i) GEOS-Chem and
(ii) WRF-Chem over (a) India, (b) China, and (c) the Middle East. The number at the top of each
panel gives the mean HFO-1234yf mixing ratios within the domains.


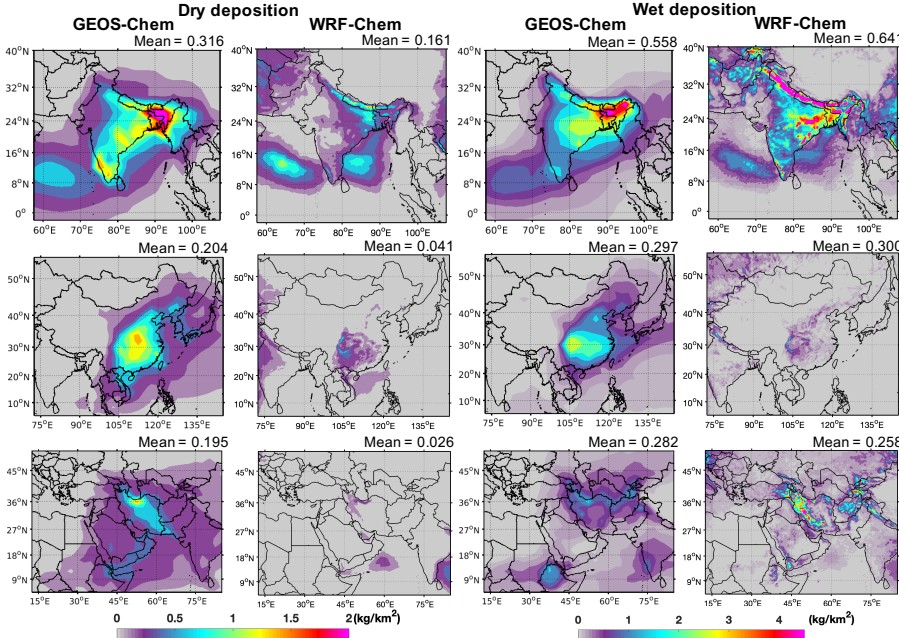


**Figure 6.** GEOS-Chem and WRF-Chem simulated annual total deposition rates of TFA (kg km$^{-2}$
yr$^{-1}$) from (a) dry and (b) wet deposition in India, China, and the Middle East domains. The number
at the top of each panel gives the mean dry and wet deposition rates within the domains.



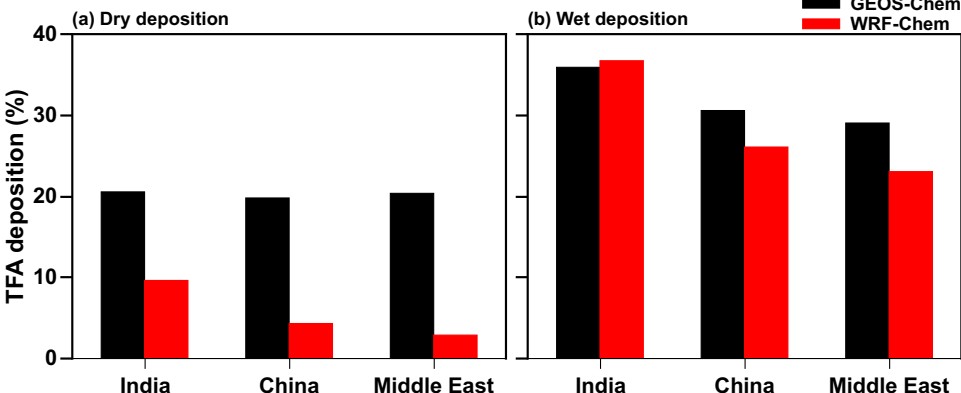

**Figure 7.** Percentage contribution of (a) dry and (b) wet deposition to total annual TFA deposition simulated in GEOS-Chem and WRF-Chem in the three domains.

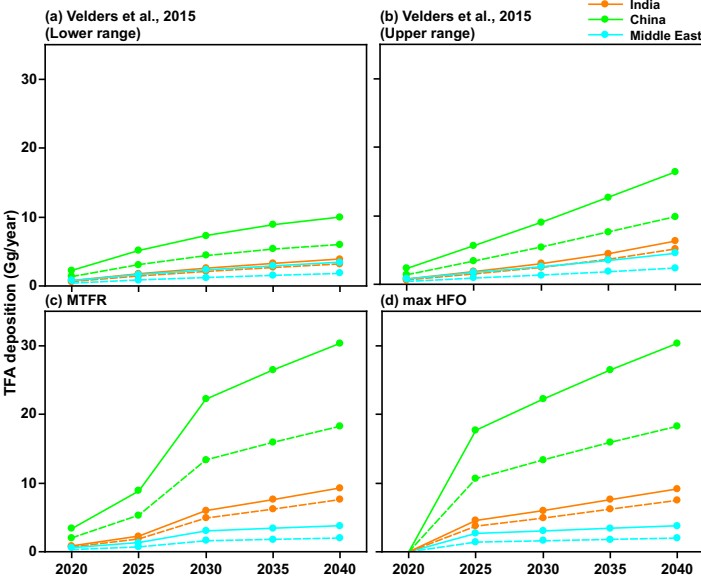

**Figure 8.** Total TFA deposited (dry and wet combined) in four emission scenarios for 2020 to 2040 within India, China, and the Middle East domains calculated using GEOS-Chem (solid lines) and WRF-Chem (dashed lines). The values from the two models are reasonably close for India and the Middle East, while the differ by almost a factor two for China.

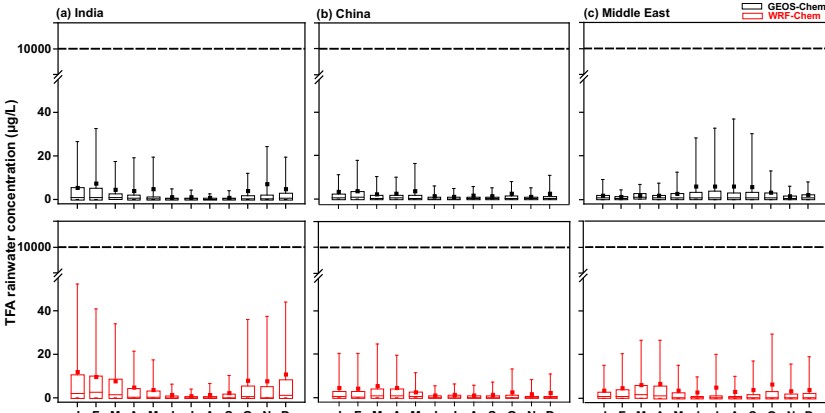

**Figure 9.** Box and whisker plot of TFA rainwater concentration calculated from GEOS-Chem and WRF-Chem in the three domains. In the box plot, the inside line and square are the median and mean, respectively. Box boundaries are 25th and 75th percentiles, and whiskers indicate the 5th and 95th percentiles. The dashed horizontal line is the No Observable Effect Concentration (NOEC) level. It is important to note that these values, including the 95th percentile values are at least 100 times lower than the NOEC for harming aquatic bodies even when normalized for higher projected emissions in 2040.

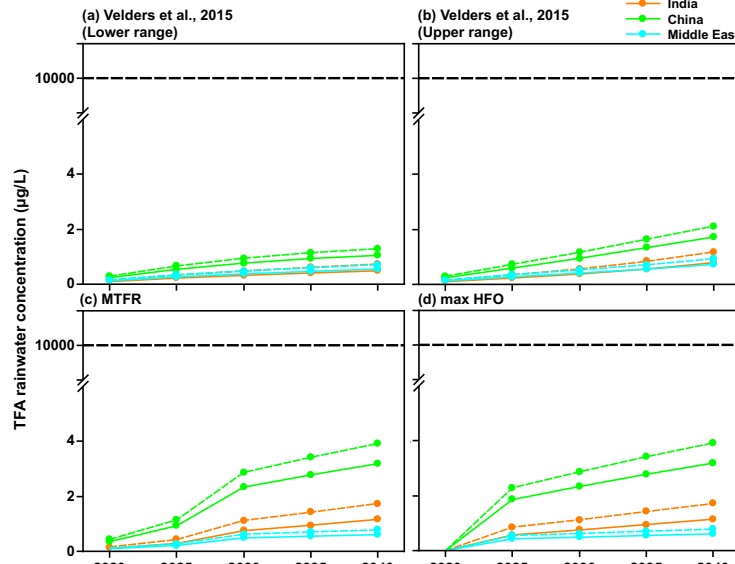

**Figure 10.** Mean TFA rainwater concentration in four scenarios for 2020 to 2040 for India, China, and the Middle East domains calculated using GEOS-Chem (solid lines) and WRF-Chem (dashed lines). The NOEC is denoted above, and it is two orders of magnitude larger than calculated TFA concentrations for any of the scenarios.




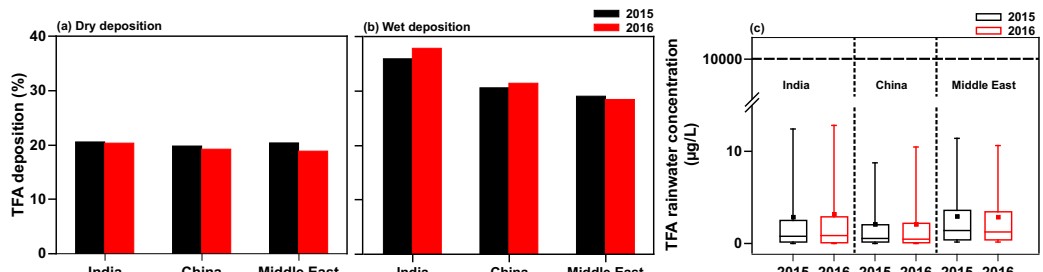


**Figure 11.** Annual percentage of total TFA (a) dry and (b) wet deposition, and (c) annual mean
TFA rainwater concentrations in India, China, and the Middle East domains from GEOS-Chem
for 2015 and 2016. The dashed horizontal line is the NOEC level.

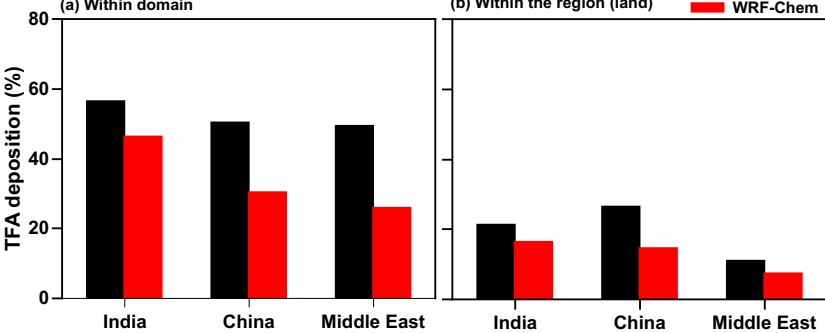

**Figure 12.** Annual percentage of total TFA deposition (dry and wet combined) calculated from
GEOS-Chem and WRF-Chem within the three (a) domains and (b) regions (land).

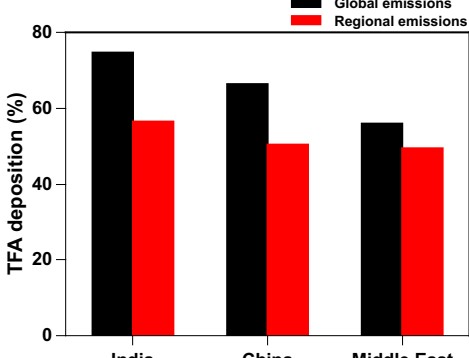

**Figure 13.** Annual percentage of total TFA deposition (dry and wet combined) in India, China,
and the Middle East from global and regional (individual regions) emissions.






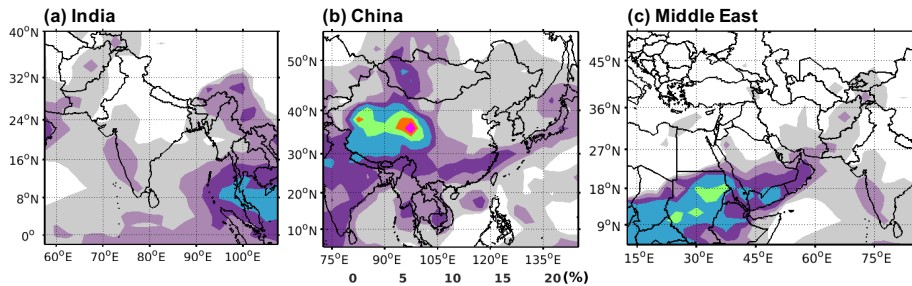


**Figure 14.** Percentage decrease in TFA deposition (dry and wet combined) by adding Criegee intermediate chemistry in (a) India, (b) China, and (c) the Middle East domains.
