# Peer review of "Trifluoroacetic acid deposition from emissions of HFO-1234yf in India, China,"

_Atmospheric Chemistry and Physics, 2021_

## Author Comment (AC1)

Atmos. Chem. Phys. Discuss., referee comment
RC1https://doi.org/10.5194/acp-2021-222-RC1,
2021

**Comment on acp-2021-222**

Anonymous Referee #1
* * *
Referee comment on "Trifluoroacetic acid deposition from emissions of HFO-1234yf in India, China and the Middle East" by Liji M. David et al., Atmos. Chem. Phys. Discuss.,https://doi.org/10.5194/acp-2021-222-RC1, 2021
* * *
This paper reports the results of a modeling study of the formation and deposition of trifluoroacetic acid (TFA) following the atmospheric oxidation of HFO-1234yf ($CF_3CF=CH_2$)emitted in India, China, and the Middle East. The concentrations of trifluoroacetic acid expected in precipitation are shown to be in a similar range to those estimated for emissions of HFO-1234yf in North America, Europe, and China (Henne et al., 2012; Kazil et al., 2014; Luecken et al., 2010; Wang et al., 2018). Similar to the previous studies modeling emissions in North America and Europe it is concluded that deposition of
trifluoroacetic acid following the atmospheric oxidation of HFO-1234yf emitted in India, China, and the Middle East has a negligible impact on human or ecosystem health. I recommend publication after the authors have considered the following minor comments.

(1) Line 56, the latest GWP estimate for HFC-134a is 1,600 (Hodnebrog et al., 2020).
The GWP value is updated in the revised manuscript (Lines 56-57).

(2) Line 65, allowance for the decomposition of chemically activated $CF_3CFHO$ radicals needs to be made in assessing the yield of $CF_3COF$ and hence trifluoroacetic acid from theatmospheric oxidation of HFC-134a (Wallington et al., 1996).  Hence, the yield of TFA from atmospheric oxidation of HFC-134a is substantially lower than 30%.
Thank you for the comment. We have added the following lines in the revised manuscript (Lines 65-68):
"Later research (Wallington et al., 1996) shows that hot $CF_3C(O)H$ formed in the degradation scheme would reduce the TFA yield from HFC-134a. This reduction is not explicitly considered here, but we acknowledge that the noted TFA yields from HFC-134a can be viewed as upper limits."

Wallington, T. J., Hurley, M. D., Fracheboud, J. M., Orlando, J. J., Tyndall, G. S., Sehested, J., Møgelberg, T. E. and Nielsen, O. J.: Role of excited CF3CFHO radicals in the atmospheric

chemistry of HFC-134a, J. Phys. Chem., 100, 18116–18122, doi:10.1021/jp9624764, 1996.

(3) The negligible impact of TFA formation during the atmospheric oxidation of HFCs, HCFCs, and HFOs has been established for some time. In 2007, the WHO Ozone report concluded "trifluoroacetic acid from the degradation of HCFCs and HFCs will not result in environmental concentrations capable of significant ecosystem damage". In 2008, Hurley et al. concluded that "the products of the atmospheric oxidation of $CF_3CF=CH_2$ have negligible environmental impact". In 2016, Solomon et al. concluded that "concentrations of TFA and its salts in the environment that result from degradation of HCFCs, HFCs, and HFOs in the atmosphere do not present a risk to humans and environment". Although some of these are cited in the latest review by Neale et al. (2021) which is cited, it would be appropriate to explicitly mention these previous reports for context.

Thank you for the comments. We have adopted the suggestion of the reviewer (verbatim!), and we have added the following lines in the revised manuscript (Lines 488-496).

"The negligible impact of TFA formation during the atmospheric oxidation of HFCs, HCFCs, and HFOs has been established for some time. The WMO/UNEP Quadrennial Ozone Layer Assessment (2007) report concluded that TFA from the degradation of HCFCs and HFCs would not result in environmental concentrations capable of significant ecosystem damage. Hurley et al., (2008) concluded in their study that the products of the atmospheric oxidation of CF3CF=CH2 will have negligible environmental impact. Solomon et al., (2016) also concluded in their study that the concentrations of TFA and its salts in the environment that result from degradation of HCFCs, HFCs, and HFOs in the atmosphere do not present a risk to humans and environment."

Hurley, M. D., Wallington, T. J., Javadi, M. S. and Nielsen, O. J.: Atmospheric chemistry of CF3CF=CH2: Products and mechanisms of Cl atom and OH radical initiated oxidation, Chem. Phys. Lett., 450, 263–267, doi:10.1016/j.cplett.2007.11.051, 2008.

Solomon, K. R., Velders, G. J. M., Wilson, S. R., Madronich, S., Longstreth, J., Aucamp, P. J. and Bornman, J. F.: Sources, fates, toxicity, and risks of trifluoroacetic acid and its salts: Relevance to substances regulated under the Montreal and Kyoto Protocols, J. Toxicol. Environ. Heal. Part B, 19(7), 289–304, doi:10.1080/10937404.2016.1175981, 2016.

WMO/UNEP Quadrennial Ozone Layer Assessment: Scientific Assessment of Ozone Depletion: 2006; WMO: Geneva, Switzerland, , 50, 572, 2007.

References

Henne, S., Shallcross, D. E., Reimann, S., Xiao, P., Brunner, D., O'Doherty, S. and Buchmann, B.: Future emissions and atmospheric fate of HFC-1234yf from mobile airconditioners in Europe, Environ. Sci. Technol., 46, 1650–1658, 2012.

Hodnebrog, Ø., Aamaas, B., Fuglestvedt, J. S., Marston, G., Myhre, G., Nielsen, C. J., Sandstad, M., Shine, K. P., and Wallington, T. J.: Updated global warming potentials andradiative efficiencies of halocarbons and related compounds, Rev. Geophys., 58, e2019RG000691, 2020.

Hurley, M. D., Wallington, T. J., Javadi, M. S., and Nielsen, O. J.: Atmospheric Chemistryof $CF_3CF=CH_2$: Products and Mechanisms of Cl Atom and OH Radical Initiated Oxidation, Chem. Phys. Lett., 450, 263-267, 2008.

Kazil, J., McKeen, S., Kim, S. W., Ahmadov, R., Grell, G. A., Talukdar, R. K. and Ravishankara, A. R.: Deposition and rainwater concentrations of trifluoroacetic acid in theUnited States from the use of Hfo-1234yf, J. Geophys. Res., 119, 14,059-14,079, 2014.

Luecken, D. J., Waterland, R. L., Papasavva, S., Taddonio, K. N., Hutzell, W. T., Rugh, J. P. and Andersen, S. O.: Ozone and TFA impacts in North America from degradation of2,3,3,3- tetrafluoropropene (HFO-1234yf), A potential greenhouse gas replacement, Environ. Sci. Technol., 44, 343–348, 2010.

Solomon, K. R., Velders, G. J. M., Wilson, S. R., Madronich, S., Longstreth, J., Aucamp, P.J., Bornman, J. F.: Sources, fates, toxicity, and risks of trifluoroacetic acid and its salts: relevance to substances regulated under the Montreal and Kyoto Protocols. J. Toxicol. Environ. Health. Part B, Crit. Rev., 19, 289-304, 2016.

Wallington, T. J., Hurley, M. D., Fracheboud, J. M., Orlando, J. J., Tyndall, G. S., Sehested,J., Møgelberg, T. E., and Nielsen, O. J.: Role of excited $CF_3CFHO$ radicals in the atmospheric chemistry of HFC-134a, J. Phys. Chem., 100, 18116-18122, 1996.

WMO (World Meteorological Organization), 2007. Scientific Assessment of Ozone Depletion: 2006, Global Ozone, Research and Monitoring Project – Report 50, Geneva,Switzerland.

---

## Author Comment (AC2)

Atmos. Chem. Phys. Discuss., referee comment
RC2 https://doi.org/10.5194/acp-2021-222-RC2,
2021

[Figure]

**Comment on acp-2021-222**

Anonymous Referee #2
* * *
Referee comment on "Trifluoroacetic acid deposition from emissions of HFO-1234yf in India, China and the Middle East" by Liji M. David et al., Atmos. Chem. Phys. Discuss., https://doi.org/10.5194/acp-2021-222-RC2, 2021
* * *
The authors present a modelling study of deposition of trifluoroacetic acid (TFA) resulting from emissions of HFO-1234f in India, China and Middle east. The HFO-1234f emissions are expected to increase in these regions in the future and such study is of significance to the atmospheric community. GEOS-Chem and WRF-Chem models were used. The models were characterized systematically, and I think the overall conclusions are scientifically sound and the paper is publishable in ACP. Below are few questions and comments I have for the authors:

-TFA rainwater concentration is dependent of the frequency of rain. I assume that the rainfall pattern will likely change in the next 20 years. Could the authors comment on how this will impact TFA rainwater concentration over the middle east region? A graphical relationship between amount of rainfall and TFA rainwater concentration for each region would be helpful for the readers.

Thank you for the comment. The variation of rainfall amounts and their geographical distribution as climate changes are uncertain. For example, Terink et al., (2013) suggest that there will be a ~20% decrease in precipitation over the Middle East region over the next 20 years. A 20% decrease in precipitation will correspond to TFA rainwater concentration still less than 40 µg/L (95$^{th}$ percentile). The annual mean precipitation over China is likely to increase, for example, by 0.078 mm/d in the 2020s and 0.218 mm/d in 2050s, with larger changes in the summer months (rainy season), according to Guo et al., (2017). The projected rainfall changes across the Indian monsoon region could increase by 6% (RCP4.5) and 8% (RCP8.5) in the mid-21$^{st}$ century (Krishnan et al., 2020). We have added some details about the future rainfall pattern in the revised manuscript (Lines 455-463):

"The variation of rainfall amounts and their geographical distribution as climate changes are uncertain, but there are some estimates. For example, Terink et al., (2013) suggest

that there could be a ~20% decrease in precipitation over the Middle East region over the next 20 years. A 20% decrease in precipitation will correspond to TFA rainwater concentration less than 40 μg/L (95th percentile). The annual mean precipitation over China is likely to increase; for example, estimates are roughly increases of 0.078 mm/d in the 2020s and 0.218 mm/d in 2050s, with larger changes in the summer months (rainy season) (Guo et al., 2017). The projected rainfall changes across the Indian monsoon region could increase by 6% (RCP4.5) and 8% (RCP8.5) in the mid-21st century (Krishnan et al., 2020)."

A graphical relationship between amount of rainfall and TFA rainwater concentration for the three regions is shown in the following figure. We have added this figure in the supplementary information (Figure S10).

[Figure]

**Figure.** Monthly mean TFA rainwater concentration (filled circles) and total precipitation (open diamonds) calculated from GEOS-Chem (solid lines) and WRF-Chem (dashed lines) over (a) India, (b) China, and (c) the Middle East domains for emissions from those regions only.

Guo, J., Huang, G., Wang, X., Li, Y. and Lin, Q.: Investigating future precipitation changes over China through a high-resolution regional climate model ensemble, Earth's Futur., 5, 285–303, doi:10.1002/2016EF000433, 2017.

Krishnan, R., Sanjay, J., Gnanaseelan, C., Mujumdar, M., Kulkarni, A. and Chakraborty, S., Eds.: Assessment of Climate Change over the Indian Region : A Report of the Ministry of Earth Sciences (MoES), Government of India, Springer Nature., 2020.

Terink, W., Immerzeel, W. W. and Droogers, P.: Climate change projections of precipitation and reference evapotranspiration for the Middle East and Northern Africa until 2050, Int. J. Climatol., 33, 3055–3072, doi:10.1002/joc.3650, 2013.

-TFA rainwater concentration in China using WRF-Chem is significantly different (factor of 2) compared with the results from GEOS-Chem and previous study by Wang et al. (shown in Figure 4). However, the percentage of wet deposition seem relatively similar in China for GEOS-Chem and WRF-Chem. Could the authors explain this difference?
Thank you for the comment. The TFA rainwater concentration in China is a factor of 2 higher in WRF-Chem because the total precipitation amounts were a factor of 1.5-2 higher in GEOS-Chem compared to WRF and TRMM (Figure S3 in the supplementary information). This difference arises because of how precipitation is dealt with in GEOS-Chem and WRF-Chem. This was one of the reasons for using two models in this study. We have added the following lines in the revised manuscript (Lines 364-366)
"The TFA rainwater concentration in China is a factor of two higher in WRF-Chem because the total precipitation amounts were factors of 1.5-2 higher in GEOS-Chem compared to WRF and TRMM as mentioned in section 3.1."

Figure 4a shows TFA deposition flux, and the differences between models can be attributed to differences in model resolutions, model transport, and precipitation. We have these details mentioned in lines 361-364.

-Recent study by Holland et al. ACS Earth Space Chem. 2021, 5, 849-857 suggest Criegee intermediates in forested region can significantly decrease the lifetime of TFA. The GEOS-Chem simulations in this study show that the overall significance is small except for the forested regions in south east Asia. I wonder if this contribution will be higher in these regions during the dry seasons.

The following figure shows the spatial pattern of the decrease in total TFA deposition (dry and wet combined) for the dry months (January-February). The overall change in TFA deposition is in the range of -2% to 7% in the three domains. In the dry months (January-February), the increase in TFA deposition due to CI was 0.03, 1.12, and 0.02 Gg (total for two months), respectively, for India, China, and the Middle East. Our study finds that the overall significance is small (less than 7%) in the forested regions even in the dry months. It is also important to distinguish between TFA produced from HFC-134a, which is much more spread out, and that from HFO-1234 yf, which is more regional.

[Figure]

**Figure.** Percentage decrease in TFA deposition (dry and wet combined) by adding Criegee intermediate chemistry over (a) India, (b) China, and (c) the Middle East domains for emissions from those regions for the dry months (January-February).

In the revised manuscript, we have added the following figure showing the percentage decrease in mean surface TFA mixing ratio by including the CI reactions with TFA after HFO-1234yf emissions from each of the regions over (a) India, (b) China, and (c) the Middle East. The decrease in the mean surface TFA mixing ratio is less than 2% (0.01 ppt) in the regions of the study.

[Figure]

**Figure.** Percentage decrease in mean (January to July) surface TFA mixing ratio by including the reaction of Criegee intermediate with TFA. The changes are shown for HFO-1234yf emissions over each of the regions: (a) India, (b) China, and (c) the Middle East domains.

We have added the following lines in the revised manuscript (Lines 589-593):
"Figure S16 shows the percentage decrease in mean surface TFA mixing ratio by including the reaction of CI with TFA following the emissions of HFO-1234yf emissions from (a) India, (b) China, and (c) the Middle East. The decrease in the mean surface TFA mixing ratio is less than 2% (0.01 ppt). Overall, the impact of CI on TFA deposition/mixing ratio was small in the regions of study."

-Is there any reason why the loss from reaction with Criegee intermediates is relatively higher in the India region in Figure 14 b) compared with 14 a)? Is this because of seasonality in rainfall?
Thank you for bringing out this point. Figure 14a corresponds to a decrease in TFA deposition (from HFO-1234yf emissions from India) by Criegee intermediate. Figure 14b corresponds to a reduction in TFA deposition (from HFO-1234yf emissions from China) by Criegee intermediates. Therefore, Figures 14a and 14b show a percentage decrease in TFA deposition by Criegee intermediates over India from HFO-1234yf emissions from two different regions. This is explicitly noted in the revised manuscript (Lines 581-583):
"Figure 14 shows the spatial pattern of decrease in total TFA deposition (dry and wet combined) for seven months by adding CI chemistry to HFO-1234yf emissions in (a) India, (b) China, and (c) the Middle East."
We have also clarified the caption for Figure 14 in the revised manuscript.

Minor comments:

-The dashes in the figures are hard to see. Please put bigger gaps in between the dashes to make them obvious.
Thank you for the comment. We have modified Figures 8 and 10 in the revised manuscript and Figures S2, S3, and S5.

-In the kinetic scheme, could the authors label the phases of all the fluorinated species. I also suggest labelling each step and then citing them in the following paragraph.
Details (phases and equation numbers) were added in the revised manuscript.

-dissociation coefficient in line 288 should have unit of mol l^-1
Corrected.

-Change 'the' to 'they' in the last line of caption for figure 8
Corrected.

-It is difficult to see some of the datapoints in Figure 9 and 10. I suggest using panel with one column and scaling the vertical axis better.
The figures have been modified in the revised manuscript.

---

## Author Response (AR2)

Dear Maria,

Thank you very much for shepherding this paper through the process. I am grateful.

I made the corrections you suggested.

In the process, I also found two typos that are now fixed:

1. on page 2 (CF3C(O)H should be CF3C(O)FH).
2. On page 2 "*Earlier research (Wallington, 1996) shows that vibrationally excited CF3CFHO formed in the degradation scheme would substantially reduce the TFA yield from HFC-134a*" should be: "==Other== *research (Wallington, 1996) shows that vibrationally excited CF3CFHO formed in the degradation scheme would substantially reduce the TFA yield from HFC-134a.*"

Thank you very much

Ravi